# Perceived Past and Current COVID-19-Stressors, Coping Strategies and Psychological Health among University Students: A Mediated-Moderated Model

**DOI:** 10.3390/ijerph191610443

**Published:** 2022-08-22

**Authors:** Maria Clelia Zurlo, Federica Vallone, Maria Francesca Cattaneo Della Volta

**Affiliations:** 1Dynamic Psychology Laboratory, Department of Political Sciences, University of Naples Federico II, 80138 Napoli, Italy; 2Department of Humanities, University of Naples Federico II, 80138 Napoli, Italy

**Keywords:** coping strategies, COVID-19 pandemic, fears, interpersonal factors, mediation-moderation, personality, psychological health, stress, time perspective, university students

## Abstract

The COVID-19-pandemic entailed abrupt/long-lasting changes in university students’ lives, resulting in growing stress and stress-related outcomes. Although nowadays the pre-pandemic-life is gradually re-establishing, past-COVID-19-stressful experiences and strategies adopted to adjust to this condition may significantly impact students’ current experiences. Therefore, the development of research exploring the complex interplay between perceived past/present COVID-19-related experiences, coping strategies, and psychological health currently reported by students is needed. This study simultaneously tests the associations between Past-COVID-19-stressors and Current-COVID-19-stressors as moderated by Coping Strategies and the associations between Past-COVID-19-stressors and Psychological Symptoms as mediated by Current-COVID-19-stressors. A mediated-moderated model was tested on 355 university students. COVID-19-Student-Stress-Questionnaire (CSSQ) was used to assess COVID-19-stressors. Two CSSQ-versions were used, one of which was adjusted to assess Past-COVID-19-stressors recalled from previous restrictive pandemic phases. Coping-Orientation-to-Problem-Experienced-New-Italian-Version and Symptom-Checklist-90-Revised were used to assess, respectively, Coping Strategies and Psychological Symptoms. Findings confirmed the hypothesized Mediated-Moderated Model. The effects of Past-COVID-19-stressors on Current-COVID-19-stressors were moderated by Coping Strategies, and the effects of Past-COVID-19-stressors on Psychological Symptoms were mediated by Current-COVID-19-stressors. Unique psychopathological risk profiles deriving from the interplay between specific past/present stressors and coping strategies were found. Researchers and clinicians can use these findings to develop updated research and timely evidence-based interventions fostering students’ adjustment in the current period. Future research should further explore the impact of the complex interplay between perceived past/present COVID-19-related experiences and individual characteristics on psychological health conditions reported by people in the aftermath of the COVID-19 pandemic.

## 1. Introduction

In the last decades, growing research attention has been given to university students’ well-being [1,2]. Indeed, the key turning point from secondary education to university life entails specific changes and stressors [3], potentially resulting in great difficulties in adapting to the new life circumstances [4,5]. A large body of studies have highlighted the multiple sources of stress which may feature university students’ experiences, arising both from the academic domain (e.g., pressures to perform; workload; time restraints; interactions with other students and academic staff) and from the non-academic life (e.g., pressures from relatives; need to achieve independence; changes in the relationships with peers/new friends/intimate partners) [3,6,7].

However, since the beginning of the COVID-19 pandemic, university students had to also deal with further changes which abruptly occurred in their life [8,9], potentially eliciting notable levels of stress related to the control measures (i.e., isolation due to the lockdown) and to the fear of being infected and infecting others, as well as to the drastic modifications in the relational and social life [10,11,12]. From this perspective, the COVID-19-related changes have deeply affected students’ interpersonal life, entailing specific experiences (i.e., loneliness; lack of connection; a frustrated sense of belonging) which expose them to a significantly high risk for psychological health [13].

Therefore, whereas several studies conducted worldwide before the pandemic have already warned about university students’ risk of reporting stress-related negative health outcomes [14,15,16], the more updated research addressing the impact of COVID-19 emergency has underlined an even higher health risk reported by students globally [17,18,19,20]. In particular, research has highlighted that university students who perceived high levels of specific COVID-19-related stressors (i.e., stress related to changes in Relationships and Academic Life, Isolation, and Fear of Contagion) were at high risk for reporting several stress-related negative outcomes, such as cognitive impairment [21], psychophysical disorders [22,23] and poor mental health [24,25]. 

Nevertheless, the presence of sources of stress does not imply the development of negative outcomes *per se*. Specifically, based on the transactional theory of stress [24,26], the latter results from two processes, namely primary appraisal (i.e., people’s assessment of perceived stressors) and secondary appraisal (i.e., people’s perceived availability of individual resources to deal with each specific stressor). Therefore, stress is a dynamic process deriving from the constant interplay between individual and situational factors, so that under the same circumstances (e.g., COVID-19-related stressors), people may report different health outcomes depending on individual differences [26]. 

Following this approach, among the wide range of individual characteristics featuring stress processes [27], coping strategies play an essential role in influencing an individual’s adjustment to situations and, thus, in determining the development of stress-related health outcomes [28,29,30]. In line with this, while most research conducted during the COVID-19 pandemic has paid great attention to the sources of stress characterizing university students ‘experiences [22,23,24,25], some studies have also explored the strategies they have adopted to deal with the new-unprecedented-reality [21,31,32,33,34].

Within the latter research direction, some studies have focused on the e-learning experience [31,32]. Indeed, a further consequence of the COVID-19 containment measures has concerned the transposition of all formal and informal activities and interactions onto online platforms, with a significant increase in the use of Information and Communication Technologies (ICTs) [35]. This has allowed the maintenance of training paths, as well as the possibility of preserving contacts and relationships with others (e.g., professors and colleagues, non-cohabiting family members, friends, and partners). On the other hand, this rapid transposition to a new “online” reality has entailed several concerns, mainly due to the increasing exhaustion connected to the daily and prolonged use of ICTs to perform all the activities (both work and leisure) and to be “connected” with others [36,37,38].

Considering research targeting stress, coping, and well-being processes among university students during the pandemic, a study conducted in Italy has highlighted that students’ higher stress levels were related to lower e-learning satisfaction [31]. This study has also provided evidence on the protective role of emotional-focused coping strategies, such as turning to religion and self-blame, suggesting the effectiveness of these strategies in dealing with the unpredictable and uncontrollable COVID-19 pandemic experiences of distance learning.

Differently, a study conducted in three European countries (Greece, Italy, and United Kingdom) has underlined that students’ recourse to problem-focused coping strategies may effectively reduce the negative effects of technology-related stressors on anxiety and depression and, conversely, the recourse to emotion-focused and avoidance-focused coping strategies may significantly exacerbate the detrimental psychological impact of technology-related stressors [32].

In the same direction, beyond the context of e-learning, research exploring stress and coping in influencing psychological health [33] and life satisfaction [34] reported by students during the pandemic have supported the protective role of problem-focused coping strategies and the negative impact of emotion-focused and avoidance-focused strategies. In addition, the well-demonstrated protective role of the recourse to positive attitude coping strategies [39,40] was also supported by research conducted during the pandemic [21,41].

However, considering the still open debate on stress process and adaptive/maladaptive coping strategies in the literature beyond the COVID-19 emergency [27,30], as well as taking into account the limited evidence on the effectiveness of coping strategies among university students during the pandemic, further research providing evidence-based and updated contributions is needed to promote students’ adjustment in the current period, namely in the crucial aftermath of the COVID-19 pandemic.

Indeed, nowadays, most of the COVID-19-related control measures are lifted, and the pre-pandemic life is gradually re-establishing, yet the pandemic has undoubtedly entailed enduring changes in university students’ lives, requiring them to employ several efforts to adjust to this unprecedented condition. From this perspective, although the current circumstances have changed (e.g., no strict rules for wearing masks, lift of social isolation), we can hypothesize that the past COVID-19-related experiences (i.e., perceived stressors) along with the resources (i.e., coping strategies) employed by students may have significant effects on their current experiences, as well as on their current psychological health conditions. The careful exploration of the moderating role of coping strategies in this complex relationship is, therefore, needed.

In line with this, previous evidence has underlined the importance of also considering the temporal dimensions when exploring the adjustment process to chronic and enduring stressful experiences and their effects on individuals’ well-being [42,43,44,45]. This suggested the need to adopt a more complex perspective of stress and coping processes so capture the psychological impact of both the current and the past—more restrictive—period. Therefore, the present study targeted the key aim to explore the complex interplay between perceived past and present COVID-19-related stress (past and current levels of stress related to changes in Relationships and Academic Life, Isolation, and Fear of Contagion) and adopted coping strategies (Problem-Solving, Positive Attitude, Social Support, Escape/Avoidance, Turning to Religion), in influencing the development of a wide set of psychological symptoms currently reported by university students (Somatization, Depression, Anxiety, Phobic Anxiety, Obsessive Compulsive, Interpersonal-Sensitivity, Hostility, Paranoid Ideation, Psychoticism).

Figure 1 shows the general conceptual mediated-moderated model depicting the relationships between perceived Past-COVID-19-stressors, perceived Current-COVID-19-stressors, adopted Coping Strategies, and self-reported current Psychological Health conditions. This proposed model tests the potential moderating role of Coping Strategies in the associations between Past-COVID-19-stressors and Current-COVID-19-stressors and, simultaneously, the potential mediating role of perceived Current-COVID-19-stressors in the associations between perceived levels of Past-COVID-19-stressors and psychological health conditions.

Therefore, the study aimed to simultaneously test the associations between Past-COVID-19-stressors and Current-COVID-19-stressors as moderated by Coping Strategies and the associations between Past-COVID-19-stressors and psychological health conditions as mediated by Current-COVID-19-stressors. Accordingly, the following hypotheses were developed and tested:

**Hypothesis** **1.***Perceived levels of Past-COVID-19-stressors are significantly associated with perceived levels of Current-COVID-19-stressors reported by university students. No prediction was made about the direction of these relationships due to the lack of evidence reported in the literature*.

**Hypothesis** **2.***Coping strategies play as significant moderators of the relationships between Past-COVID-19-stressors and Current-COVID-19-stressors reported by university students. No prediction was made about the direction of these relationships due to the still limited and mixed evidence reported in the literature*.

**Hypothesis** **3.***Perceived levels of Current-COVID-19-stressors play a significant mediator in the relationships between Past-COVID-19-stressors and psychological health conditions currently reported by university students*.

**Hypothesis** **4.***The mediated-moderated path models tested show a good fit of data*.

## 2. Materials and Methods

This observational cross-sectional study was carried out on a sample of Italian university students. From April 2022 to June 2022, students were invited to participate in an online survey by using institutional channels (e.g., mailing lists) as well as via informal channels (e.g., social media). Participation was voluntary and students received no rewards. All the information about the research project and about the privacy policy (e.g., the treatment and the confidentiality of their data) were included within the online form. The research was carried out in line with the 1964 Helsinki declaration and its later amendments/comparable ethical standards, and it was approved by the Ethical Committee [Masked for Blind Revision] (IRB:14/2022). Overall, 355 university students (Women *n* = 248, 63.4%; Age *M* = 20.84, *SD* = 2.97) provided the informed consent and completed the survey. The survey consisted of the following questionnaires: the COVID-19-Student-Stress-Questionnaire (CSSQ), the Coping-Orientation-to-Problem-Experienced-New-Italian-Version (COPE-NIV), and the Symptom-Checklist-90-Revised (SCL-90-R).

### 2.1. COVID-19-Student Stress Questionnaire

The COVID-19 Student Stress Questionnaire (CSSQ) [12] was used to measure perceived levels of COVID-19-related stressors reported by university students. It consists of 7 items on a 5-point Likert scale ranging from zero, “Not at all stressful,” to four, “Extremely stressful.” The CSSQ is divided into three subscales, namely Relationships and Academic Life (4 items), Isolation (2 items), and Fear of Contagion (single item). A Global Stress score (range = 0–28; Cronbach’s α = 0.71) was also provided. In the original version, students were asked to complete the survey by reflecting on their experiences during the current period of the COVID-19 pandemic. In the present study, two alternative CSSQ-versions (i.e., instructions) were used, i.e., the original one was used to assess perceived levels of Current-COVID-19-stressors, whereas another one was adjusted to assess Past-COVID-19-stressors. For the latter version, students were asked to recall their past experiences, referring to the previous restrictive pandemic phases.

### 2.2. Coping Orientation to Problem Experienced–New Italian Version

The Coping Orientation to Problem Experienced–New Italian Version (COPE [46]; COPE-NIV Italian version [47]) was used to measure coping strategies adopted by students. It consists of 60 items on a five-point Likert scale ranging from one “I usually don’t do this at all” to four “I usually do this a lot.” The COPE-NIV is divided into five subscales, namely Problem-Solving (12 items; Cronbach’s α = 0.83); Positive Attitude (12 items; Cronbach’s α = 0.76); Escape/Avoidance (16 items; Cronbach’s α = 0.70); Social Support (12 items; Cronbach’s α = 0.88); Turning to Religion (8 items; Cronbach’s α = 0.85).

### 2.3. Symptom-Checklist-90-Revised

The Symptom Checklist-90-Revised (SCL-90-R [48]; Italian version [49]) was used to measure psychological health conditions currently reported by university students. It consists of 90 items on a 5-point Likert scale ranging from zero “Not at all” to four “Extremely”. The SCL-90-R is divided into nine subscales, namely Somatization (12 items; Cronbach’s α = 0.83), Obsessive-Compulsive (10 items; Cronbach’s α = 0.82), Interpersonal Sensitivity (9 items; Cronbach’s α = 0.83), Depression (13 items; Cronbach’s α = 0.87), Anxiety (10 items; Cronbach’s α = 0.84), Phobic-Anxiety (7 items; Cronbach’s α = 0.68), Hostility (6 items; Cronbach’s α = 0.80), Paranoid Ideation (6 items; Cronbach’s α = 0.76), Psychoticism (10 items; Cronbach’s α = 0.77). A global score, namely the Global Severity Index (GSI; 90 items; Cronbach’s α = 0.97), was also provided. It indicates both the number of symptoms and the intensity of the psychological disease.

### 2.4. Statistical Analysis

Descriptive statistics and intercorrelations between the study variables (Spearman’s correlations) were conducted by using SPSS (Version 21). Furthermore, referring to the hypothesized theoretical model (Figure 1), the mediated-moderated models (i.e., each for every unique interplay between stressors, coping strategies, and psychological health outcomes) were tested by using Amos Graphics (Version 21). However, before testing each specific hypothesized model, we have preliminarily tested our general framework by using the global indices scores of Global Stress (from the CSSQ) and Global Severity Index (from the SCL-90-R). Specifically, a mediated-moderated model including Past and Current COVID-19-Global Stress (main and mediating effects), all coping strategies (main and moderating effects), as well as Global Severity Index (outcome), was drawn and tested. Afterward, in line with the transactional approach [26], specific mediated-moderated models were tested to achieve tailored information on the effects of the unique interplay between specific sources of stress (Relationships and Academic Life; Isolation; Fear of Contagion), coping strategies (Social Support; Problem-Solving; Positive Attitude; Turning to Religion; Avoiding) and psychological health conditions (Somatization, Depression, Anxiety, Phobic Anxiety, Obsessive Compulsive, Interpersonal-Sensitivity, Hostility, Paranoid Ideation, Psychoticism). Specifically, the associations between Past-COVID-19-stressors and Current-COVID-19-stressors (Hypothesis 1) as moderated by Coping Strategies (Hypothesis 2) and, simultaneously, the associations between Past-COVID-19-stressors and psychological health conditions as mediated by Current-COVID-19-stressors (Hypothesis 3) were explored. Standardized regression coefficients were provided along the paths of each model and the associations were considered significant at *p* < 0.05. The bootstrap-based SEM was used [50]. Indeed, Bootstrapping is a great technique suggested to manage the potential non-normality distribution of data in SEM, and it can provide a convenient way of checking the robustness of data and gaining more accurate estimates [51,52,53,54,55]. Overall, the models fit were tested by using standard goodness-of-fit indices: χ^2^ non-significant (*p* > 0.05), Goodness-of-Fit (GFI > 0.90), Tucker–Lewis Index (TLI > 0.95), Comparative Fit Index (CFI > 0.95), and Root Mean Square Error of Approximation (RMSEA < 0.08) (Hypothesis 4).

## 3. Results

Descriptive statistics and intercorrelations between the study variables are displayed in Table 1.

Preliminarily, the associations between perceived levels of Past- and Current-COVID-19-stressors (including Global Stress) were explored (Hypothesis 1). Data revealed that Past-COVID-19-stressors were significantly positively associated with Current-COVID-19-stressors reported by university students (*p* < 0.01). Likewise, Past-Global Stress and Current-Global Stress were found to be significantly positively related (*p* < 0.01). These data supported the validity of testing the proposed mediated-moderated models.

Afterward, before testing (Hypotheses 2–4), a general mediated-moderated model including Past and Current COVID-19-Global Stress (main and mediating effects), all coping strategies (main and moderating effects), as well as Global Severity Index (outcome) was preliminarily examined. The statistically significant mediated-moderated model which emerged (Figure 2) included the following dimensions: Past- and Current- Global Stress; Avoiding, Problem Solving, Positive Attitude, and Social Support Coping Strategies; Global Severity Index. Turning to Religion Coping Strategy was removed from the general model since non-significant (*p* > 0.05). This was done due to the necessity to keep parsimony in complex statistical models. Indeed, non-significant parameters can be considered unimportant to the model and, in the interest of scientific parsimony, should be deleted [56].

Data revealed that Avoiding, Problem Solving, Positive Attitude, and Social Support Coping Strategies significantly moderated the associations between Past- and Current- Global Stress, whereas Current-Global Stress significantly mediated the association between Past- Global Stress and Global Severity Index. The path model showed good fit (χ^2^ = 110.22, GFI = 0.95, TLI = 0.96, CFI = 0.97, and RMSEA = 0.05).

Considering (Hypotheses 2–4), specific mediated-moderated path models emerged as statistically significant and are illustrated according to the specific COVID-19-stressors: Relationships and Academic Life (Figure 3, Figure 4 and Figure 5); Isolation (Figure 6, Figure 7 and Figure 8); Fear of Contagion (Figure 9, Figure 10 and Figure 11).

Specifically, considering the significant models for Relationships and Academic Life, the Avoiding Coping Strategy significantly moderated the associations between Past- and Current-Relationships and Academic Life stressors (Hypothesis 2), whereas Current Relationships and Academic Life stressors mediated the association between Past-Relationships and Academic Life and Anxiety (Hypothesis 3). The path model showed good fit (χ^2^ = 111.06, GFI = 0.99, TLI = 0.97, CFI = 0.99, and RMSEA = 0.03) (Hypothesis 4) (Figure 3).

Secondly, the Avoiding Coping Strategy also significantly moderated the associations between Past- and Current-Relationships and Academic Life stressors (Hypothesis 2), whereas Current Relationships and Academic Life stressors mediated the association between Past-Relationships and Academic Life and Somatization (Hypothesis 3). The path model showed good fit (χ^2^ = 109.04, GFI = 0.91, TLI = 0.96, CFI = 0.97, and RMSEA = 0.05) (Hypothesis 4) (Figure 4).

Finally, Problem Solving Coping Strategy significantly moderated the associations between Past- and Current-Relationships and Academic Life stressors (Hypothesis 2), whereas Current-Relationships and Academic Life stressors mediated the association between Past-Relationships and Academic Life and Anxiety (Hypothesis 3). The path model showed good fit (χ^2^ = 107.02, GFI = 0.98, TLI = 0.97, CFI = 0.96, and RMSEA = 0.05) (Hypothesis 4) (Figure 5).

Considering the significant models for Isolation, Social Support Coping Strategy significantly moderated the associations between Past- and Current-Isolation stressors (Hypothesis 2), whereas the Current-Isolation stressor mediated the association between Past-Isolation stressor and Depression (Hypothesis 3). The path model showed good fit (χ^2^ = 104.07, GFI = 0.92, TLI = 0.96, CFI = 0.98, and RMSEA = 0.04) (Hypothesis 4) (Figure 6).

Moreover, Positive Attitude Coping Strategy significantly moderated the associations between Past- and Current- Isolation stressors (Hypothesis 2), whereas the Current Isolation stressor mediated the association between Past-Isolation stressors and Depression (Hypothesis 3). The path model showed good fit (χ^2^ = 108.26, GFI = 0.94, TLI = 0.97, CFI = 0.95, and RMSEA = 0.05) (Hypothesis 4) (Figure 7).

Finally, Avoiding Coping Strategy significantly moderated the associations between Past- and Current-Isolation stressors (Hypothesis 2), whereas the Current-Isolation stressor mediated the association between the Past-Isolation stressor and Interpersonal-Sensitivity (Hypothesis 3). The path model showed good fit (χ^2^ = 111.46, GFI = 0.99, TLI = 0.98, CFI = 0.97, and RMSEA = 0.03) (Hypothesis 4) (Figure 8).

Considering the significant models for Fear of Contagion, Avoiding Coping Strategy significantly moderated the associations between Past- and Current-Fear of Contagion stressors (Hypothesis 2), whereas Current-Fear of Contagion stressor mediated the association between Past-Fear of Contagion stressor and Phobic-Anxiety (Hypothesis 3). The path model showed good fit (χ^2^ = 108.43, GFI = 0.90, TLI = 0.95, CFI = 0.95, and RMSEA = 0.05) (Hypothesis 4) (Figure 9).

Moreover, the Avoiding Coping Strategy also significantly moderated the associations between Past- and Current-Fear of Contagion stressors (Hypothesis 2), whereas Current-Fear of Contagion stressors mediated the association between Past- Fear of Contagion stressor and Psychoticism (Hypothesis 3). The path model showed good fit (χ^2^ = 103.76, GFI = 0.94, TLI = 0.96, CFI = 0.97, and RMSEA = 0.05) (Hypothesis 4) (Figure 10).

Finally, Problem Solving Coping Strategy significantly moderated the associations between Past- and Current-Fear of Contagion stressors (Hypothesis 2), whereas the Current-Fear of Contagion stressor mediated the association between Past-Fear of Contagion stressor and Obsessive-Compulsive symptoms (Hypothesis 3). The path model showed good fit (χ^2^ = 112.08, GFI = 0.98, TLI = 0.95, CFI = 0.96, and RMSEA = 0.05) (Hypothesis 4) (Figure 11).

## 4. Discussion

The present study targeted university students and aimed to simultaneously test the associations between Past-COVID-19-stressors and Current-COVID-19-stressors as moderated by Coping Strategies and the associations between Past-COVID-19-stressors and psychological health conditions as mediated by Current-COVID-19-stressors. In this direction, the study aimed to provide updated contributions to develop tailored and evidence-based interventions promoting students’ adjustment in the current crucial period, namely in the aftermath of the COVID-19 pandemic.

Firstly, according to the transactional theory of stress [26], one key result of this study concerns the significant associations between perceived levels of Past-COVID-19-stressors and Current-COVID-19-stressors. Indeed, in line with research on the detrimental effects of past stressful events [42,43,44,45,47], our findings supported the hypothesis that the subjective assessment of the current pandemic period as being “highly stressful” may be influenced by past stressful experiences. In this direction, our data have demonstrated that long-lasting exposure to significant challenges has led to higher students’ vulnerability in the current period.

However, the main result of the present study concerns evidence supporting the validity of the proposed conceptual mediated-moderated model depicting the relationships between perceived Past and Current COVID-19-stress (both Global Stress and specific stressors), Coping Strategies, and self-reported current psychological health conditions (both Global Severity Index and specific symptoms). Indeed, to the best of our knowledge, this is the first study applying the transactional approach [26] to demonstrate—by using a time perspective—the effects of the complex interplay between past/current COVID 19 stressors and coping strategies on psychological health reported by university students in the current time of the COVID-19 pandemic.

Nonetheless, despite its theoretical implications, several practical implications can be derived from the unique psychopathological risk profiles resulting from the interplay between specific past/present stressors and coping strategies. Specifically, considering perceived levels of stress related to changes in Relationships and Academic Life, data revealed that students’ recourse to Avoiding Coping Strategy exacerbated the negative effects of perceived Past-COVID-19-stressors on Current-COVID-19-stressors, while students’ recourse to Problem-Solving Coping Strategy was able to buffer this association significantly. These findings are in line with research [32,33,34], highlighting both the detrimental effects of the recourse to strategies centered on avoiding coping and—conversely—the positive role played by strategies centered on problem-solving. However, these findings may also help to identify students who might be potentially at high risk due to their attempts to deal with the overwhelming changes in relational, academic, and social life by withdrawing/escaping negative feelings. In this direction, these findings suggested the need to promote more active strategies as well as to support thorough reflections on past experiences to prevent mental disease escalation. Indeed, data also revealed that Current perceived levels of stress related to Relationships and Academic Life mediated the association between Past-Relationships and Academic Life and, respectively, Anxiety and Somatization. These findings provided a greater understanding of the underlying pathways of the relationship between past experiences and current psychological health through present COVID-19-related experiences. However, these findings also suggested that the intimate interplay between our study variables may determine specific psychological symptoms beyond the overall impact on psychological health (i.e., Global Severity Index from SCL-90-R). In the case of high levels of stress related to Relationships and Academic Life, for example, data highlighted the risk for reporting anxiety and its physical manifestations (somatic anxiety). The latter symptomatology could also be interpreted considering the present abrupt shift from online learning to the return, after a long-lasting period, to face-to-face university life. This, indeed, requires—once again—adjustment efforts (e.g., face-to-face exams; relationships with professors and colleagues). The latter phenomenon may be especially true for those students who started university life with online learning (never attended in presence university and experienced university life). Therefore, they may—even more—feel a lack of connection with others and a “frustrated sense of belonging” to social groups and community (peers and academic community) [13], so requiring the development of timely interventions and prevention strategies in this complex transition period.

Considering perceived stress related to Isolation, the negative impact of avoiding coping was confirmed, whereas more complex moderating roles of, respectively, social support coping and positive attitude were found. Specifically, although the recourse to social support emerged as a potential resource (in itself), it was however not able to significantly buffer the negative impact of perceived stress related to Past-isolation on perceived stress related to Current-isolation. This finding was in line with previous evidence suggesting the potential negative impact of social support coping [57,58], mainly when the source of support is—indeed—unavailable (i.e., in our case, due to the COVID-19-related control measures and lockdown). From this perspective, students who were used to dealing with stress by adopting social support strategies may have felt highly frustrated due to the impossibility of accessing their sources of support (friends, relatives, partners) beyond the use of ICTs [37,59]. This may have resulted in exacerbation of feelings of demoralization, loneliness, isolation, and thwarted belongingness, which is well-demonstrated to be extremely harmful to individuals’ mental health [13,60,61,62].

In this direction, starting from the beginning of the COVID-19 emergency, the scientific and clinical communities had warned about the idea that the extreme experiences related to the pandemic would have a detrimental impact on interpersonal/social well-being and psychological health (i.e., vicious cycle), requiring people to employ several adjustment resources [63].

Considering students, nowadays, two major issues they must handle: 1. the need to re-establishing a satisfactory interpersonal life (sense of belonging, face-to-face connections, balance in the use of ICTs) [64,65]; 2. the need to deal with the economic crisis and its consequences effectively [63,66]. However, these two goals are intimately linked as research has highlighted that perceived satisfaction in social/relational life and interpersonal trust were key factors that significantly preserve mental health even during economic crises [67,68,69]. Accordingly, the abovementioned research suggested the need to develop timely measures and interventions (i.e., by policymakers, Higher Education administrations, and health professionals) to support university students in re-establishing fundamental social and interpersonal networks, starting from academic life.

Furthermore, data revealed that students who adopted positive attitude coping strategies, who were, indeed, able to reappraise the extreme condition of lockdowns effectively and social distancing reported lower levels of current-perceived stress due to isolation. Therefore, in line with research on the protective role of positive attitude coping [21,39,40,41], these students seemed to be able to effectively re-adjust to the Current-changed-circumstances. Once again, this is by gradually restoring the possibility to benefit from their relational and social network. Conversely, those who reported currently high stress related to isolation also reported symptoms of Interpersonal-Sensitivity (low self-esteem; high self-doubt; feeling of being uncomfortable in interpersonal relationships) and depression. We hypothesize that these students are probably still actually isolated, and they are still experiencing remarkable difficulties in interacting with people and feelings of loneliness. These findings should be carefully considered also in light of the well-demonstrated depressive risk registered among university students both in the pre-pandemic period [70,71,72,73] and during the pandemic [32,74,75,76], as well as considering the alarming suicidal risk linked to disruptions within interpersonal relations [13,59,61,62,63].

Finally, considering perceived stress related to Fear of Contagion, findings supported—once again—the protective role of problem-solving coping. Nevertheless, data also highlighted the protective role of avoiding coping strategies in buffering the negative impact of perceived levels of stress related to Past-Fear of Contagion on perceived levels of stress related to Current-Fear of Contagion. These findings were in contrast with the research trend on the maladaptive role of the recourse to avoiding coping [32,33,34] but were, instead, in line with a growing branch of research emphasizing that active avoiding/distracting coping may also play as a resource under some circumstances [30,77,78,79]. In this perspective, students’ possibility to distance themselves from the potentially paralyzing fear of being infected and infecting loved ones may have resulted in a better adjustment to the current condition. A recent study conducted among the general population [80] has explored the wide set of COVID-19-related fears (both during the lockdown and post-lockdown). This study [80] has underlined the remarkable presence of fears linked not only to isolation and loneliness but also those fear of dying/getting sick, as well as of being stigmatized (in case of infection or suspicion of illness). Surprisingly, this study also revealed that fears of being infected or dying from COVID-19 are more frequently reported by younger rather than elderly people [80], indicating the need to counteract the potential risk of underestimating the impact of perceived fear of contagion among the student population. In this direction, our findings have also suggested that students who still feel stress related to fear of contagion were at higher risk of reporting Phobic-Anxiety, Obsessive-Compulsive symptoms, and even Psychoticism, also linked to the prolonged detachment to reality and the consequent biased view of the current condition. This also considers the advancement in medical control over the COVID-19 virus (i.e., population-wide vaccination campaign; new tailored drugs; fewer death rates, and more no-symptomatic people, although the still high number of new cases). Moreover, these results should be carefully considered also in light of the research warning about the risk of reporting phobic anxiety and obsessive-compulsive symptoms [81,82,83], as well as psychotic symptoms linked to the pandemic experiences [84,85].

Overall, findings underlined the notable risk for students who passively escaped from facing past COVID-19-related feelings/experiences and attempted to rely on unavailable sources of social support during lockdowns. They, indeed, may still report high stress related to changes in relational and university life and isolation. Nonetheless, the possibility to actively face and re-appraise past experiences (i.e., by using problem-solving and positive attitude coping) and, to a certain extent, to adjust to the new—less terrifying—reality by distancing from the fear of contagion may result in higher levels of well-being reported by university students in the current period.

From this perspective, in line with the transactional approach [26], when students fail to deal with the past effectively and current stressors, these complex stress-and-coping processes may result in a remarkable risk for reporting: 1. Anxiety and Somatization linked to perceived stress related to changes in Relationships and Academic Life; 2. Interpersonal-Sensitivity and Depression linked to perceived stress related to Isolation: 3. Phobic-Anxiety, Obsessive-Compulsive and Psychoticism linked to perceived stress related to Fear of Contagion. Interventions should mainly target students reporting these emerged risk profiles.

Despite the key findings derived from the present study, some limitations should be addressed. Firstly, the study is based on a convenience sample of university students, which may have limited the generalizability of the results. In the same direction, despite its merits (e.g., lowered research costs, reaching the target populations during the COVID-19 pandemic), the use of online surveys may have entailed the risk of selection bias related to the recruitment. Therefore, further investigations on larger and more representative samples of the student population from Italy are needed to allow the generalizability of these results (e.g., nationally representative sample, more men; paper-pencil administering). The validity of these results was also limited by the adoption of a cross-sectional design, the use of self-report measures, and the inclusion of retrospective data (i.e., recall of events). In particular, the recall method—mainly for the reconstruction of previous perceived levels of stress—may provide less accurate, complete, and consistent data (e.g., the current psychological health condition may have biased the memory of the experience; students may overestimate/underestimate their present/past experiences and their current symptoms). However, considering the low retention interval (experiences to be recalled occurred within two years) and the aim of assessing perceived levels of stress and psychological health (rather than objective data, for example, by using parameters such as salivary cortisol levels), we believe that using the recall method might have affected the accuracy of our findings to a little extent. Despite this, future research could be developed with a longitudinal design and also include objective assessment measures. This would reduce the risk of bias and allow for obtaining more consistent data. In the same direction, students were not asked if they have started/continued a psychological treatment during the two years of the pandemic, so it is possible there have been changes (e.g., use of more adaptive strategies to deal with stressors) we are not able to capture with this set of data. Nevertheless, in line with the transactional perspective [26], coping represents a personality characteristic (so relatively stable over time and less amenable to sudden changes and modifications). Therefore, we consider that our findings on the associations between coping and both past- and current- perceived stressors could be affected by the lack of this information to a small extent. Finally, despite the complexity of the tested framework, future research could also include further individual characteristics (e.g., socio-demographic characteristics), mainly other personality characteristics such as personality traits [60] which may have a significant role in influencing the stress-and-health process among university students.

## 5. Conclusions

Despite these limitations, the study has several implications. From the theoretical perspective, the study provided a valid mediated-moderated model, addressing the complex interplay between perceived past and present COVID-19-related sources of stress and adopted coping strategies to predict the psychological health conditions of university students after two years of COVID-19 emergency.

Moreover, several practical implications could be highlighted. Indeed, both researchers and clinicians should consider the meaningfulness of assessing—by using tailored and valid tools (e.g., CSSQ)—individuals’ perceived levels of COVID-19-related stressors, exploring both their current and past experiences.

In the same direction, researchers and clinicians should consider the need to assess a wide range of psychological symptoms, including—but not limited to—anxiety and depression, to achieve a more realistic and comprehensive portrait of the psychological impact of the COVID-19 pandemic.

Moreover, considering data emerged on the complex role of coping strategies, interventions should reduce students’ tendency to deal with COVID-19 challenges linked to the relational/academic/social life by avoiding/detaching from negative feelings. This by promoting, instead, different strategies (i.e., problem-solving and positive attitude) to actively face past and current experiences. However, interventions should also consider the potential protective role of the recourse, to a small extent, to avoiding/distancing strategies when dealing with the fear of contagion, so preventing the escalation/chronicization of phobic-anxious and obsessive-compulsive symptoms and, eventually, psychotic symptoms. Finally, considering that students who used to adopt social support coping may have experienced—for more than two years—the frustrations linked to the unavailability of these resources (due to the lockdowns), our study highlighted the need to foster within tailored interventions, a careful evaluation of the possibility to benefit, once again, of the relational and social domains to deal with stressors.

In conclusion, this study provided a valid framework to develop evidence-based interventions fostering university students’ adjustment and psychological health in the current crucial transition period of the pandemic.

## Figures and Tables

**Figure 1 ijerph-19-10443-f001:**
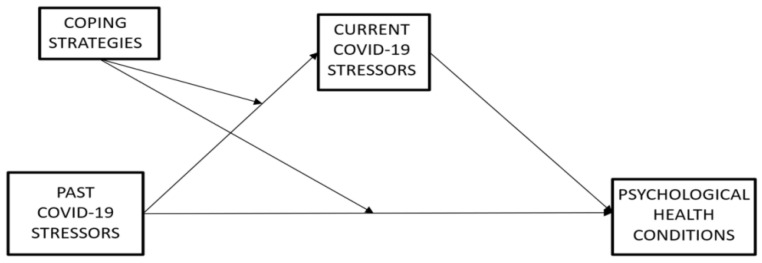
The conceptual mediated-moderated model.

**Figure 2 ijerph-19-10443-f002:**
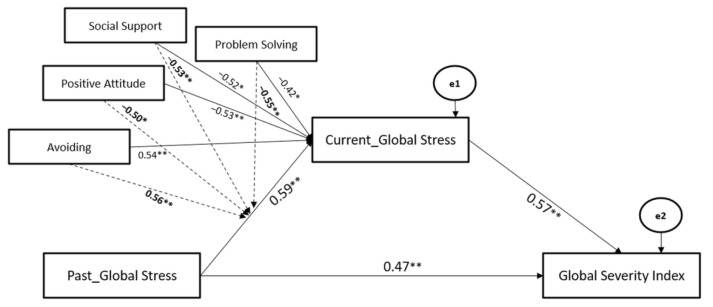
The Mediated-Moderated model of Past and Current Global Stress, Coping Strategies, and Global Severity Index. Moderating effects values are reported in bold along the dotted arrows. ** *p* < 0.01; * *p* < 0.05.

**Figure 3 ijerph-19-10443-f003:**
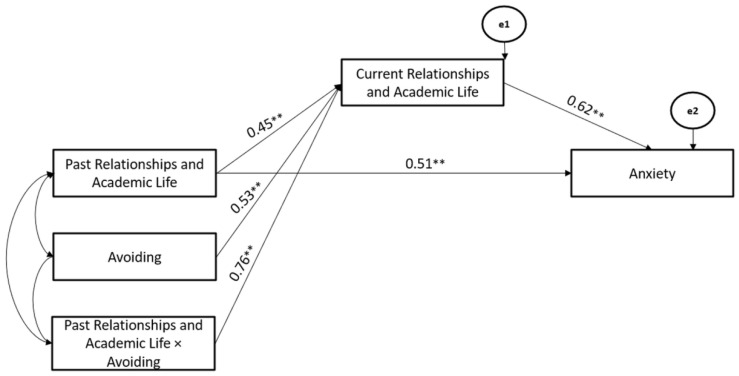
The Mediated-Moderated model of Past and Current Relationships and Academic Life stressors, Avoiding Coping Strategy, and Anxiety. ** *p* < 0.01.

**Figure 4 ijerph-19-10443-f004:**
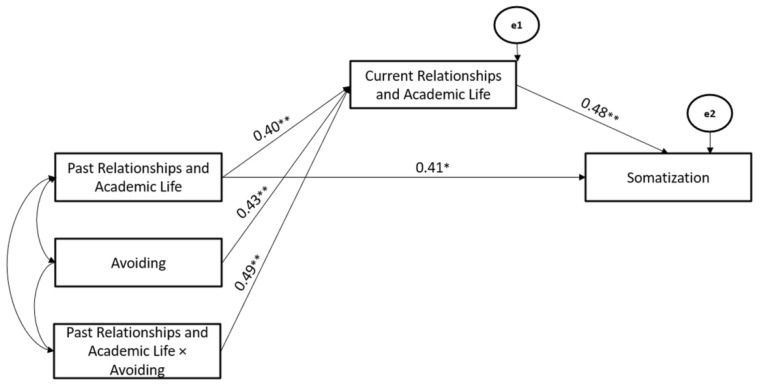
The Mediated-Moderated model of Past and Current Relationships and Academic Life stressors, Avoiding Coping Strategy, and Somatization. ** *p* < 0.01; * *p* < 0.05.

**Figure 5 ijerph-19-10443-f005:**
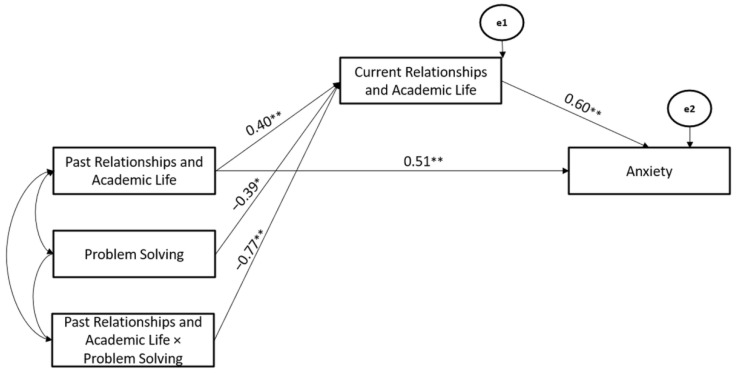
The Mediated-Moderated model of Past and Current Relationships and Academic Life stressors, Problem Solving Coping Strategy, and Anxiety. ** *p* < 0.01; * *p* < 0.05.

**Figure 6 ijerph-19-10443-f006:**
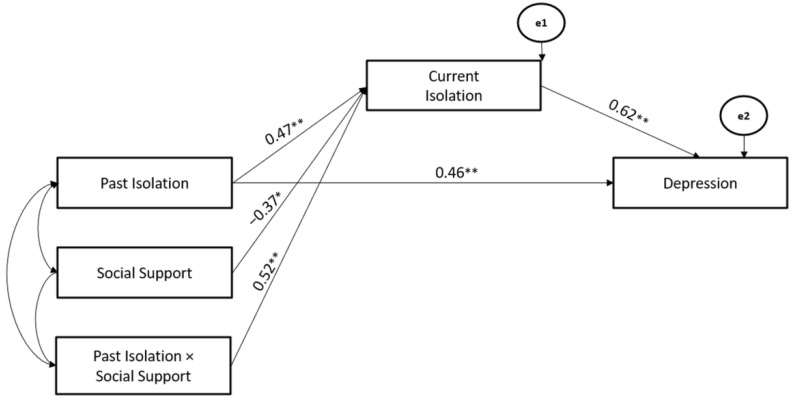
The Mediated-Moderated model of Past and Current Isolation stressors, Social Support Coping Strategy, and Depression. ** *p* < 0.01; * *p* < 0.05.

**Figure 7 ijerph-19-10443-f007:**
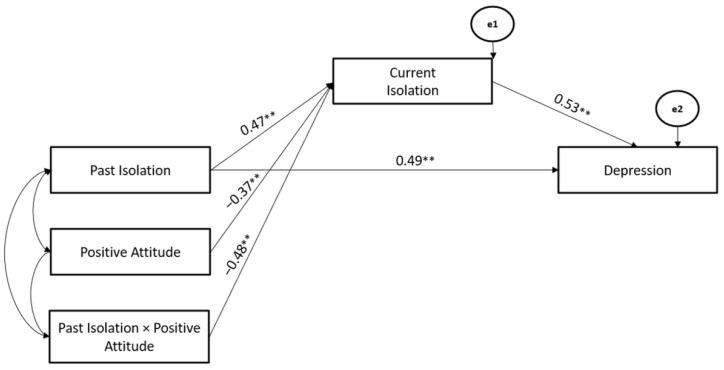
The Mediated-Moderated model of Past and Current Isolation stressors, Positive Attitude Coping Strategy, and Depression. ** *p* < 0.01.

**Figure 8 ijerph-19-10443-f008:**
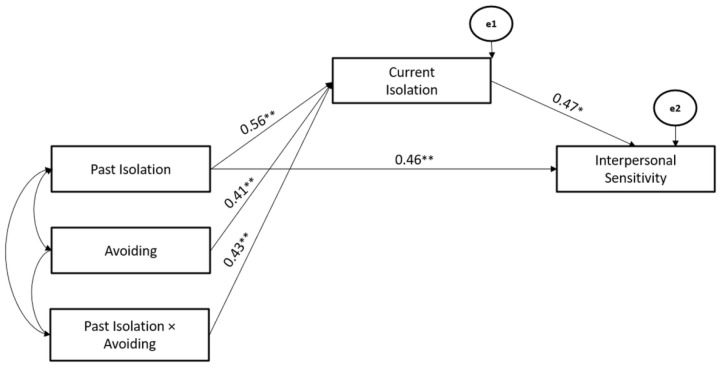
The Mediated-Moderated model of Past and Current Isolation stressors, Avoiding Coping Strategy, and Interpersonal Sensitivity. ** *p* < 0.01; * *p* < 0.05.

**Figure 9 ijerph-19-10443-f009:**
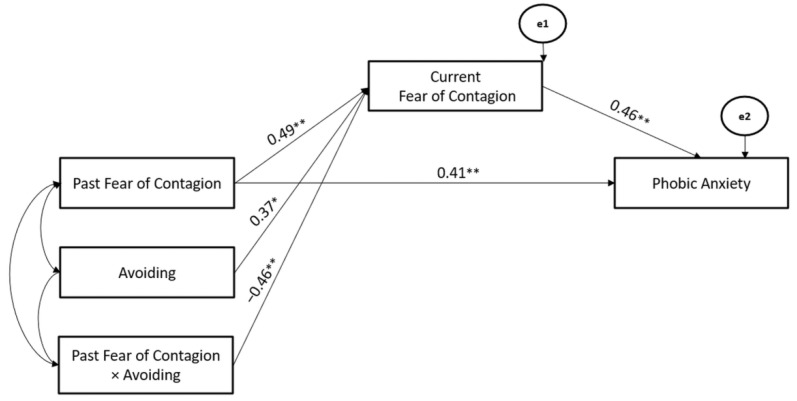
The Mediated-Moderated model of Past and Current Fear of Contagion stressors, Avoiding Coping Strategy, and Phobic Anxiety. ** *p* < 0.01; * *p* < 0.05.

**Figure 10 ijerph-19-10443-f010:**
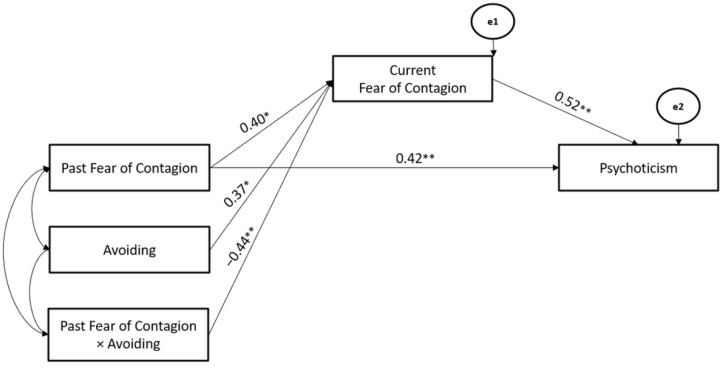
The Mediated-Moderated model of Past and Current Fear of Contagion stressors, Avoiding Coping Strategy, and Psychoticism. ** *p* < 0.01; * *p* < 0.05.

**Figure 11 ijerph-19-10443-f011:**
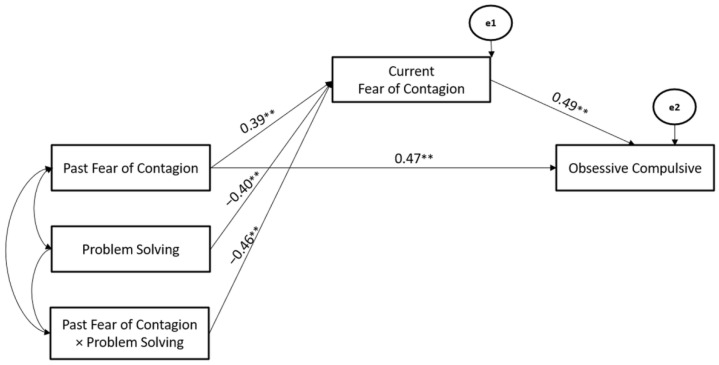
The Mediated-Moderated model of Past and Current Fear of Contagion stressors, Problem Solving Coping Strategy, and Obsessive-Compulsive symptoms. ** *p* < 0.01.

**Table 1 ijerph-19-10443-t001:** Means, Standard Deviations, and Intercorrelations between study variables.

	*M*	*SD*	1	2	3	4	5	6	7	8	9	10	11	12	13	14	15	16	17	18	19	20	21	22	23
**COVID-19-STRESSORS**																									
1. Past-Relationships and Academic Life	7.34	3.47	1																						
2. Past-Isolation	4.99	2.19	0.46 **	1																					
3. Past-Fear of Contagion	2.40	1.12	0.33 **	0.38 **	1																				
4. Past-Global Stress	14.77	5.47	0.89 **	0.77 **	0.56 **	1																			
5. Current- Relationships and Academic Life	4.80	3.31	0.49 **	0.28 **	0.29 **	0.49 **	1																		
6. Current-Isolation	2.61	2.07	0.30 **	0.35 **	0.25 **	0.39 **	0.46 **	1																	
7. Current-Fear of Contagion	1.23	1.09	0.14 *	0.10	0.51 **	0.24 **	0.42 **	0.45 **	1																
8. Current-Global Stress	8.62	5.38	0.43 **	0.32 **	0.37 **	0.48 **	0.89 **	0.76 **	0.63 **	1															
**COPING STRATEGIES**																									
9. Social Support	31.04	8.51	0.11	0.18 *	0.20 **	0.19 *	0.05	0.11	0.08	0.10	1														
10. Problem Solving	30.36	5.95	0.03	0.07	0.01	0.04	−0.09	0.01	−0.06	−0.08	0.13	1													
11. Positive Attitude	31.66	5.36	0.19 *	0.19 **	0.06	0.21 *	−0.02	−0.06	−0.11	−0.08	0.12	0.51 **	1												
12. Turning to Religion	17.02	5.13	0.02	−0.04	0.12	0.03	0.16 *	0.08	0.24 **	0.17 *	−0.06	0.04	0.23 **	1											
13. Avoiding	29.26	7.19	0.18 *	0.25 **	0.08	0.21 **	0.31 **	0.10	0.04	0.25 **	0.15 *	−0.15 *	0.07	−0.22 **	1										
**PSYCHOLOGICAL HEALTH**																									
14. Somatization	1.29	0.97	0.33 **	0.22 **	0.18 *	0.35 **	0.43 **	0.24 **	0.19 **	0.40 **	0.02	−0.07	−0.04	0.01	0.39 **	1									
15. Obsessive-Compulsive	1.59	0.86	0.36 **	0.19 **	0.17 *	0.34 **	0.47 **	0.24 **	0.14 *	0.41 **	0.11	−0.11	−0.06	−0.04	0.49 **	0.74 **	1								
16. Interpersonal-Sensitivity	1.39	0.86	0.36 **	0.19 **	0.23 **	0.33 **	0.46 **	0.24 **	0.212 **	0.40 **	0.03	−0.11	−0.15 *	0.01	0.37 **	0.65 **	0.72 **	1							
17. Depression	1.60	0.91	0.42 **	0.21 **	0.17 *	0.38 **	0.47 **	0.24 **	0.17 *	0.41 **	0.05	−0.14	−0.07	−0.08	0.41 **	0.76 **	0.82 **	0.83 **	1						
18. Anxiety	1.39	0.96	0.34 **	0.26 **	0.21 **	0.36 **	0.41 **	0.20 **	0.15 *	0.36 **	0.05	−0.03	−0.07	−0.05	0.43 **	0.84 **	0.82 **	0.74 **	0.86 **	1					
19. Hostility	1.14	0.92	0.20 **	0.22 **	0.08	0.21 **	0.32 **	0.16 *	0.06	0.27 **	0.07	−0.07	−0.15*	−0.09	0.47 **	0.65 **	0.65 **	0.64 **	0.67 **	0.69 **	1				
20. Phobic-Anxiety	0.69	0.77	0.25 **	0.19 **	0.28 **	0.29 **	0.46 **	0.24 **	0.32 **	0.44 **	0.09	−0.13	−0.17*	0.04	0.34 **	0.61 **	0.56 **	0.65 **	0.61 **	0.59 **	0.51 **	1			
21. Paranoid Ideation	1.35	0.90	0.28 **	0.19 **	0.16 *	0.28 **	0.35 **	0.26 **	0.15 *	0.34 **	0.05	−0.09	−0.07	−0.07	0.48 **	0.67 **	0.71 **	0.73 **	0.73 **	0.72 **	0.68 **	0.56 **	1		
22. Psychoticism	1.02	0.80	0.32 **	0.21 **	0.18 *	0.31 **	0.44 **	0.21 **	0.16 *	0.38 **	−0.04	−0.12	−0.10	−0.01	0.52 **	0.69 **	0.75 **	0.80 **	0.77 **	0.75 **	0.73 **	0.63 **	0.81 **	1	
23. Global Severity Index	1.28	0.78	0.41 **	0.28 **	0.23 **	0.40 **	0.50 **	0.25 **	0.17 *	0.43 **	0.05	−0.15	−0.10	−0.08	0.50 **	0.87 **	0.89 **	0.87 **	0.93 **	0.93 **	0.78 **	0.70 **	0.84 **	0.88 **	1

** *p* < 0.01; * *p* < 0.05.

## Data Availability

The data that support the results of this study are available from the corresponding author upon reasonable request.

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
