# Peer review of "Perceived Past and Current COVID-19-Stressors, Coping Strategies and Psychological Health among University Students: A Mediated-Moderated Model"

_ijerph, 2022, doi:10.3390/ijerph191610443_

Round 1

Reviewer 1 Report

This is well written manuscript, with comprehensive literature review, sound Methods and statistical analysis, and in-depth discussion.

One question from me is that, while people’s coping strategies may change over time, available support resources, as well as their specific circumstance, how authors would justify their associations of current coping strategies with past-COVID-stress?

A Global Stress score was mentioned in the Methods. Authors could consider specifying the role of this score in the study, as a criterion reference?

The potential information bias of retrospective data collection on past-COVID-stress, together with the impacts of results, deserves a more thorough discussion.

Another two minor changes:

1.       Implications for practice and future research to be included in Abstract.

2.       Typo: P2 line 72, the “In line with this, whether…” should be “In line with this, while…”

Reviewer 2 Report

The manuscript presents evidence on how students’ past and present covid experience and coping strategies may impact on students’ current psychological health. The topic is of value and findings should be of wide interest in both academic circles and to the public.  

There are a few issues, mostly in the statistical approaches that I would like the authors to address more fully to ensure findings are robust.

1. Past covid experience is measured as recall.  Could respondents’ recall be affected by their current psychological condition?  Can you defend against that? If not, how might it affect the study findings?

 2. Statistical Analysis and results

a)       Please extend the statistical analysis section (and revise the language in figure 1 if necessary) to explain estimation of your empirical models more fully and relate the estimations performed to the results presented in the SEM diagrams. The strategy for developing and testing models by including separate subsets of variables is unclear.

 In figure 1, the use of plurals “psychological health conditions”, current and past “stressors” and “coping strategies” suggests that one complex model will be estimated in which all conditions, stressors and strategies are modelled simultaneously. In contrast the results show separate models for one health condition, one current and past stressor and one coping strategy.  It is possible that underlying mechanisms include more than one of each simultaneously, and omissions can produce biased results.

For example in figures 2 and 4 both models correspond to the outcome Anxiety, the current and past stressor in both is “Relationships and Academic life” but they differ in coping strategies. Similarly Figures 5 and 6 differ only in coping strategies.  Why not model each outcome including multiple (or all) stressors and coping strategies simultaneously? Or if the focus is on each stressor (in line with the ordering of results) why not include more than one coping strategy simultaneously, and possibly more than one health outcome. Omission of relevant variables can lead to biased estimates.

b) Use of Pearson’s correlation coeff.  Pearson’s correlation coefficient should not be computed with Likert scales particularly created from a limited number of items (eg Fear of Contagion – single item, Isolation – 2 items.)

c)       Explain briefly your choice to use Boot-strapping in the SEM. Is it to overcome problems of non-normality?

Reviewer 3 Report

Thank you.

Best regards.
